# Benefits of *Chlorella vulgaris* against Cadmium Chloride-Induced Hepatic and Renal Toxicities via Restoring the Cellular Redox Homeostasis and Modulating Nrf2 and NF-KB Pathways in Male Rats

**DOI:** 10.3390/biomedicines11092414

**Published:** 2023-08-29

**Authors:** Mayada R. Farag, Mahmoud Alagawany, Eman A. A. Mahdy, Enas El-Hady, Shimaa M. Abou-Zeid, Suzan A. Mawed, Mahmoud M. Azzam, Giuseppe Crescenzo, Azza M. A. Abo-Elmaaty

**Affiliations:** 1Forensic Medicine and Toxicology Department, Veterinary Medicine Faculty, Zagazig University, Zagazig 44519, Egypt; 2Poultry Department, Faculty of Agriculture, Zagazig University, Zagazig 44519, Egypt; 3Anatomy and Embryology Department, Veterinary Medicine Faculty, Zagazig University, Zagazig 44519, Egypt; dremanmahdy82@gmail.com (E.A.A.M.); dr.enas.anatomy@gmail.com (E.E.-H.); 4Department of Forensic Medicine and Toxicology, Faculty of Veterinary Medicine, University of Sadat City, Sadat City 6012201, Egypt; shima10_a@yahoo.com; 5Zoology Department, Faculty of Science, Zagazig University, Zagazig 44519, Egypt; zoologist_zoologist@yahoo.com; 6Department of Animal Production, College of Food & Agriculture Sciences, King Saud University, Riyadh 11451, Saudi Arabia; mazzam@ksu.edu.sa; 7Department of Veterinary Medicine, University of Bari “Aldo Moro”, Valenzano, 70010 Bari, Italy; giuseppe.crescenzo@uniba.it; 8Pharmacology Department, Faculty of Veterinary Medicine, Zagazig University, Zagazig 44519, Egypt; azza8885@yahoo.com

**Keywords:** CdCL_2_, *Chlorella vulgaris*, oxidative stress, liver, kidney, NF-kB, rats

## Abstract

In our life scenarios, we are involuntarily exposed to many heavy metals that are well-distributed in water, food, and air and have adverse health effects on animals and humans. Cadmium (Cd) is one of the most toxic 10 chemicals reported by The World Health Organization (WHO), affecting organ structure and function. In our present study, we use one of the green microalga *Chlorella vulgaris* (ChV, 500 mg/kg body weight) to investigate the beneficial effects against CdCl_2_-induced hepato-renal toxicity (Cd, 2 mg/kg body weight for 10 days) on adult male Sprague-Dawley rats. In brief, 40 adult male rats were divided into four groups (*n* = 10); Control, ChV, Cd, and Cd + ChV. Cadmium alters liver and kidney architecture and disturbs the cellular signaling cascade, resulting in loss of body weight, alteration of the hematological picture, and increased ALT, AST, ALP, and urea in the blood serum. Moreover, cadmium puts hepatic and renal cells under oxidative stress due to the up-regulation of lipid peroxidation resulting in a significant increase in the IgG level as an innate immunity protection and induction of the pro-inflammatory cytokines (IL-1β and TNF-α) that causes hepatic hemorrhage, irregular hepatocytes in the liver and focal glomeruli swelling and proximal tubular degeneration in the kidney. ChV additive to CdCl_2_, could organize the protein translation process via NF-kB/Nrf2 pathways to prevent oxidative damage by maintaining cellular redox homeostasis and improving the survival of and tolerance of cells against oxidative damage caused by cadmium. The present study shed light on the anti-inflammatory and antioxidative properties of *Chlorella vulgaris* that suppress the toxicity influence of CdCl_2_.

## 1. Introduction

The diverse health influence of exposure to environmental heavy metals became a global issue due to their toxic effects on living organisms, either by metabolic interference or mutagenesis [1]. Cadmium (Cd) has been reported to be among the highly toxic metals entering the water supply and contaminating food or air from different industrial and agricultural sources [2]. Long-term exposure to Cd has been found to cause variable physiological, histological, and biochemical alterations to different body organs of humans and laboratory animals with slow elimination from the body. Hence, they accumulate mainly in the liver, kidney, and brain [3,4,5,6]. Moreover, acute Cd exposure caused hepatotoxicity and death in different living species [7,8,9,10].

Furthermore, various in vitro and in vivo research revealed that Cd preferentially accumulated in the hepatocytes resulted in enhancement of lipid peroxidation and alteration of mitochondrial function [11,12]. Alongside, Cd resulted in genotoxicity and apoptotic effects in the human hepatocytes [13]. Sufficient evidence has emerged to reveal the mechanisms of hepatotoxicity induced by Cd, including ischemia initiated by endothelial damages and activation of Kupffer cells that induce serial cascades of events, including the proinflammatory and cytotoxic mediator, and activation of Nrf2 pathway that induce oxidative stress via suppression of antioxidant related genes [14,15,16].

Additionally, many studies revealed that Cd could induce renal injury via synthesizing the Cd-metallothionein (Cd-Mt) complex inside the hepatic tissues which circulated into blood, reaching the proximal renal tubules [17,18,19]. The free Cd-ions might lead to free radicals accumulation, oxidative stress, and lipid peroxidation [20,21]. Generally, oxidative stress might adversely affect the structure and physiological function of cells and impair the translation and transcription of RNA besides the structures and functions of cell membranes [22]. Furthermore, it has been investigated that metabolic alteration, hormonal disorders, and secretion of pro-inflammatory biomarkers have also been observed in association with Cd-induced oxidative damage [23,24]. Additionally, the reactive oxygen species (ROS) control the activity of heat shock protein70 (HSP70), nuclear factor erythroid 2-related factor 2 (Nrf2) and nuclear factor kappa B (NF-kB), and heme oxygenase-1 (HO-1), which could regulate the antioxidant responses of the cells in oxidative stress conditions. Nrf2 and NF-kB significantly enhance the antioxidant defense of stressed cells and help to eliminate and detoxify exogenous chemicals and their toxic metabolites [25].

*Chlorella vulgaris* (*C. vulgaris*) is a single-cell, easily cultivated, highly productive green microalga most often used in food supplements owing to its high content of valuable nutrients [26,27]. Accordingly, the Food and Drug Administration (FDA) documented *C. vulgaris* as a safe alga for the dietary supplement [28]. It is a superfood, containing 18% amino acids, 20% vitamins, 60% protein, and essential elements such as calcium, iron, magnesium, phosphorous, and potassium [29]. Furthermore, microalgae have valuable antioxidants and therapeutic properties as they contain carotenoids, chlorophyll, phycobiliproteins, lutein, and astaxanthin [30]. *Chlorella* sp. supplementation has many beneficial physiological properties as they are antihypertensive [31], antioxidative [32], hypocholesterolemic [33], have antitumor activities [34], and hypolipidemic and hypoglycemic roles in human and animal research [35].

Furthermore, *C. vulgaris* has been reported to protect against hepatic injuries induced by carbon tetrachloride in mice and rats and is highly resistant to toxic heavy metals, including cadmium [36,37]. Moreover, the detox efficacy of *C. vulgaris* was also reported against CCl_4_, which causes renal damage, including glomerulosclerosis, atrophy, and loss of cellular entity in the renal cortex [38]. The hepatorenal protective effects of *C. vulgaris* against cadmium toxicity in rats were reported [3,4]. Perhaps the supposed mechanisms of detoxification in the presence of *C. vulgaris* are illustrated through the inductions of metal binding metallothionein MT-like protein in the cell, which is involved in heavy metal detoxification, and once the toxic elements fixed to the *C. vulgaris* fibers, which could not be reabsorbed, they are eliminated from the body in the stool [39,40]. Recently, *C. vulgaris* has been reported to safely remove pesticides, heavy metals, and herbicides from the bodies by increasing their excretion in urine and feces [41]. However, scientific information concerning the exact mode of action and molecular mechanisms of *C. vulgaris* on the toxicological effects of cadmium is relatively little. Therefore, this work aimed to investigate the benefits of *C. vulgaris* supplementation in cadmium detoxification by evaluating function markers of the liver and kidney, immunological markers, biomarkers of oxidative stress, stress hormones, stress-responsive genes mRNA expression, and histopathological changes in the liver and kidney of rat.

## 2. Materials and Methods

### 2.1. Animal Husbandry

Adult male rats (Sprague–Dawley; *n* = 40), with 180–190 gm average body weight (BW), were used in our study. Rats were housed under hygienic conditions in stainless-steel cages with wood shavings as bedding and acclimatized before use for two weeks on a basal diet, with free access to water, with 12 h light-darkness cycle.

### 2.2. Tested Compounds

CdCl_2_ (analytical grade with 99% purity) was brought by El-Faraana Company for trading in Giza, Cairo, Egypt, as a white powder. *Chlorella vulgaris* was kindly supplied by The National Research Center, Giza, Cairo, Egypt. The rest of the chemicals were obtained from Sigma Chemical Co. (Sigma-Aldrich, Inc., PO Box 14508, St. Louis, MI, USA) and were of analytical grade.

### 2.3. Experimental Design

After two weeks of acclimatization, the 40 rats were divided randomly into 4 equal groups, each containing 10 rats; group 1 (control group). Group 2 is the ChV group (received *Chlorella vulgaris* at a dose of 500 mg/kg body weight/day orally for 10 days). Group 3 (treated with 2 mg/kg CdCl_2_ via subcutaneous injection daily for 10 days) and group 4 (Cd + ChV) co-treated with both 2 mg/kg CdCl_2_ and 500 mg/kg *Chlorella vulgaris* daily for 10 days with the same indicated routes.

### 2.4. Blood Collecting and Tissue Preservation

At the experimental end (after 10 days), rats were weighed and then euthanized by cervical dislocation under anesthesia using intramuscular injection with 1 mL/kg of ketamine xylazine mixture (2:1). For each experimental group, samples of blood (5 samples per group) were harvested from the median canthus in an anticoagulant-free tube for separation of serum to be used for biochemical analysis. Another set of blood (5 samples per group) was collected in tubes containing EDTA for hematological analysis. For tissue collection, dissected liver and kidney specimens were weighed to obtain the relative weight (organ weight ×100/body weight) and then divided into 3 parts; the 1st part was centrifuged at 4 °C for 15 min at 3000 rpm to obtain a homogenate, then the supernatants were harvested and kept at −20 °C to be used in biochemical analysis. The 2nd part was kept at −80 °C for quantitative real-time qRT-PCR, and the 3rd specimens were fixed in 10% neutral buffered formalin for the histopathological procedure.

### 2.5. Hematological Analysis

Collected blood samples were used to measure hematologic parameters by blood cell analyzer. Total blood picture, white blood cell count, and differential leukocyte count were indicated using an automatic cell counter (Hospitex Hemascreen 18, Hospitex International CO, Via Baldanzese, Calenzano, Italy) [42].

### 2.6. Biochemical Analysis

After serum collection (5 samples/group) was used to determine the total protein [43], albumin [44], total cholesterol [45], high-density lipoprotein (HDL) low-density lipoprotein (LDL) [46], triglycerides [47] creatinine, urea, glucose were estimated spectrophotometrically using commercial diagnostic kits purchased from Biodiagnostic Company, Giza, Egypt. Alkaline phosphatase (ALP), aspartate aminotransferase (AST), and alanine aminotransferase (ALT) were estimated spectrophotometrically using commercial diagnostic kits purchased from Biodiagnostic Company (Giza, Egypt) as described in [48,49], respectively, according to the manufacturer’s instructions. The serum levels of immunoglobulin (IgG and IgM) were measured as described in [50] using commercial ELISA kits for rats (CusabioBiotech Co., Ltd., Wuhan, China) according to the manufacturer’s protocols.

### 2.7. Inflammatory Markers Evaluation in the Serum

Inflammation parameters, including Interleukin-1 beta (IL-1β), tumor necrosis factor-α (TNF-α), and nitric oxide (NO) were estimated [51] using commercial rat ELISA kits (Catalog no: MBS825017, MBS843321, MBS010567, respectively, from MyBioSource, San Diego, CA, USA) following the manufacturer’s directives.

### 2.8. Oxidative Stress Markers Detection in the Serum

Oxidative stress is indicated by determining reactive oxygen species (ROS) and total antioxidant capacity (TAC) [52] by the ELISA kit (Cat No. MBS164653, MyBioSource, San Diego, CA, USA for ROS and TA 2513 kit, Biodiagnostic Co., Egypt for TAC). Additionally, total antioxidant capacity was detected using commercial kits. While malondialdehyde; MDA (lipid peroxidation marker) [53] was estimated using commercial kits (Cat No. ab118970, Abcam Co., Cambridge, UK) according to the manufacturer’s instructions.

### 2.9. Determination of Stress-Related Hormones

Cortisone, adrenaline, and noradrenaline were evaluated in the serum by rat ELISA kits (Cat. no: MBS024061, MBS031232, and MBS269993, respectively, from MyBioSource, San Diego, CA, USA) according to the instructions of the manufacturer [54].

### 2.10. Separation of mRNA and Quantitative Real-Time PCR (qRT-PCR)

RNA was extracted from the liver tissues, and its concentration and integrity were checked by agarose (1%) and spectrophotometry. QuantiTect RT kit (Qiagen, Germany) synthesized first-strand cDNA. The primers of the targeted genes (*HSP70*, Nrf2, NF-kB, HO-1, and the internal housekeeping gene *β-actin*) [55,56,57] are shown in Appendix A. QuantiTect SYBR Green PCR kit (Qiagen, Germany) and a Rotor-Gene Q apparatus Real-time were used for performing the PCR. The thermocycler condition was 95 °C for 15 min for the initial activation, followed by 40 cycles of denaturation at 94 °C for 15 s, annealing at 60 °C for 10 s, and elongation at 72 °C for 15 s. The relative expressions of the targeted genes were analyzed using the 2^−ΔΔCt^ equation [58].

### 2.11. Histopathological Investigation

Dissected kidney and liver samples were fixed for 48 h in 10% neutral-buffered formalin. Then samples were washed carefully under running tap water for one night, followed by dehydration in ascending ethanol series (70–100%). After that, tissues were cleared in xylene and embedded in paraffin wax. Paraffin blocks were cut at 3–5 µm thickness, stained with H&E (hematoxylin and eosin) stain, and examined by Olympus BX51 Light Microscope, Tokyo, Japan [59].

### 2.12. Statistical Analysis

Data were analyzed using One-way ANOVA followed by Duncan’s Multiple Range test to compare means value between groups. Data were expressed as mean ± standard error (SE), and a *p* < 0.05 was considered significant.

## 3. Results

### 3.1. Chlorella vulgaris Improved the Hematological Parameters Altered by Cadmium

Table 1 shows the hematological parameters after Cd and *C. vulgaris* treatment. The RBC count was not significantly altered in the different experimental groups; however, PCV and Hb declined significantly after Cd administration. Interestingly, co-exposure to *C. vulgaris* in the presence of Cd could elevate the Hb and PCV levels and restore Hb to the normal value. Furthermore, the results of the leukogram revealed that the total leukocyte count was significantly decreased with a concomitant reduction in the lymphocyte count compared with the control group after Cd treatment. Contrarily, co-administration of *C. vulgaris* + Cd restored the lymphocytic count to the control value and significantly elevated the WBC count, though it was still below normal level. While no significant differences were found in the neutrophils and eosinophils counts among the different experimental groups.

### 3.2. C. vulgaris Restores the Body and Organs Weight Affected by Cd Treatment

After Cadmium administration, the body weight of treated rats showed a significant reduction compared with the control and *C. vulgaris* groups. Adding *Chlorella vulgaris* could restore the final weight of exposed rats in the Cd + *C. vulgaris* group to the normal level. However, the different treatments did not significantly alter the relative weight of the kidneys, liver, and spleen (Table 2).

### 3.3. C. vulgaris Modulates the Serum Biochemical Parameters Altered by Cd Treatment

Rats injected with cadmium exhibited a remarkable decline in serum levels of albumin and total protein (*p* < 0.001) relative to the control. Conversely, *C. vulgaris +* Cd elevated the total protein and albumin values without returning them to the normal level (Table 3). ALT, AST, ALP, and urea levels were upregulated significantly (*p* < 0.001) in the serum of the cadmium-treated group relative to the control. Interestingly, *C. vulgaris* addition to Cd significantly decreased their levels, unlike Cd alone. Moreover, data also showed a significant elevation in the creatinine level in Cd-treated rats relative to the control; however, its level in the *C. vulgaris +* Cd group was comparable to the control.

Concerning the lipid profile, rats exposed to Cd showed significant elevations in the LDL-cholesterol and total cholesterol levels, accompanied by a significant reduction in the HDL-cholesterol level in serum compared to the control. Contrarily, the co-exposure to *C. vulgaris +* Cd significantly lowered the LDL-cholesterol and total cholesterol levels and elevated the HDL-cholesterol compared to the Cd group. However, the different treatments could not restore the serum triglycerides to the normal value (Table 3).

### 3.4. Cd Administration Induces the Whole Body’s Oxidative Stress

The level of oxidative stress and TAC were evaluated in the serum of treated groups. Herein, the MDA total and ROS were increased significantly by Cd (*p* ˂ 0.01) compared to the control. Co-administration of *C. vulgaris +* Cd modulated the level of MDA to normal value. Moreover, *C. vulgaris +* Cd group exhibited a significant reduction in ROS level; however, it did not normalize to the control level. Collectively, the findings indicated that the lowest level of TAC was observed in the Cd-treated group, followed by *C. vulgaris +* Cd; while the highest values were observed in the *C. vulgaris* group compared to the control (Table 4).

### 3.5. Cd Administration Elevates the Innate Immunity Response

Data in Table 4 indicated a significant increase in the level of IgG in the Cd group relative to the control; however, its level in the *C. vulgaris +* Cd group was comparable to the control. IgM levels did not significantly differ among the different groups.

### 3.6. Cd Administration Stimulates the Pro-Inflammatory Mediators

Regarding the inflammatory response, Cd significantly increased the serum levels of TNF-α and IL-1β, which were significantly suppressed (*p* ˂ 0.01) by the co-administration of *C. vulgaris* but did not reach to normal value (Table 4).

### 3.7. C. vulgaris Additive Modulates the Adrenal Hormones Induced by Cd Administration

Cd induced significant elevations in the levels of stress-related hormones, including cortisone, noradrenaline, and adrenaline, compared to the control (Figure 1). However, their levels exhibited no remarkable changes in the *C. vulgaris* group compared to the control one. Interestingly, these hormones modulated again in the *C. vulgaris* + Cd co-treated group to the control value.

### 3.8. Chlorella vulgaris Restores the Cellular Redox Homeostasis by Modulating Stress Key Mediators at the Genetic Level

Cellular key mediators control the expressions of various genes shared in cellular antioxidant, antitoxic, and anti-inflammatory responses. Some of these mediators include the NF-KB, 70 kDa heat shock protein (Hsp70), Nrf2, and HO-1 [60,61]. Herein, mRNA expressions of hepatic HSP70 and NF-kB involved in the cellular mediation against Cd toxicity exhibited significant up-regulations compared to the control and *C. vulgaris* groups. Co-administration of *C. vulgaris +* Cd significantly down-regulated the relative expressions of both genes compared with the Cd group (Figure 2A,B).

Conversely, treatment with Cd showed significant decreases in the relative expression of HO-1 and Nrf-2 mRNAs compared to the control and *C. vulgaris* groups. Contrarily, the co-administration of *C. vulgaris +* Cd significantly improved the Nrf-2 mRNAs expression than Cd alone. Similarly, HO-1 expression was up-regulated significantly. *Chlorella* v. restored Nrf-2 and Ho-1 expressions to the control value (Figure 2C,D).

### 3.9. C. vulgaris Relief the Toxicity of CADMIUM in the Hepatic and Renal Tissues at the Histological Levels

At the histological level, the liver of the control as well as *C. vulgaris* groups showed normal hepatic architecture with central veins and hepatocyte cords in the portal area (Figure 3A,B). Contrarily, Cadmium causes alternation in the hepatic tissue architecture exhibited by hepatocyte degeneration, inflammatory cellular infiltration, and blood congestion in the portal area (Figure 3C). Interestingly, *C. vulgaris* inhibited the cadmium-induced inflammatory pathway and reduced the hepatic tissue damage (Figure 3D) not only in the liver but also the renal corpuscles of the kidney as the control and *C. vulgaris* groups exhibited normal renal cortex with Bowman’s capsule containing glomerulus and fine-arranged proximal and distal convoluted tubules with central arranged nuclei in the lining epithelia (Figure 4A,B). In the cadmium-treated group, histopathological alternation in the renal cortex was observed in segmented glomeruli, dilated tubules, edema exudate, and congested blood vessels (Figure 4C). Interestingly, adding *C. vulgaris* to the cadmium in the co-exposed group ameliorated the renal tissue with fine-arranged proximal and distal convoluted tubules (Figure 4D).

## 4. Discussion

Cadmium is a highly toxic metal that is widely distributed in the surrounding environment with a destructive impact on many organ systems [62,63,64,65]. In vitro studies revealed that ROS production underlined cadmium’s toxicity mechanisms because cadmium induces the production of nitric oxide, superoxide anion, and hydrogen peroxidase, which is implicated in many deleterious health effects [66,67,68].

Furthermore, hematological profiles are considered good indicators to evaluate the physiological response of the animal to internal or external stressors [69]. In our present study, the administration of CdCl_2_ significantly decreased Hb, PVC, WBC count, and lymphocytes (which are immune cells fundamental in cellular and humoral immunity. In the blood, they represent 20 to 45% of WBC), and this agreed with earlier studies which reported that CdCl_2_ could alter the hematological indices [70,71]. Additionally, Cd induced a remarkable reduction in the total body weight of rats. However, the relative weight of the liver and kidneys were not significantly altered by different treatments, Similar findings were observed by other authors [71,72]. *C. vulgaris* in our present study could improve the altered hematological parameters and normalize the Cd-reducing effect on body weight.

*C. vulgaris* is rich in pigments (chlorophyll), amino acids, vitamins (A, B complex, C, and E), and minerals (iron, calcium, manganese, phosphorus) [73,74]. Accordingly, its content of antioxidant vitamins could protect cells against the unwanted actions of free radicals [75]. Vitamin E could protect cellular membranes against lipid peroxidation by scavenging lipid peroxyl radicals, counteract oxidative damage, and keep the ascorbic acid and GSH contents in damaged cells after exposure to xenobiotics, including Cd [76,77]. Moreover, *C. vulgaris* has been used as a useful food additive for fish diets as it could improve immunity, enhance digestibility, and organize growth performance owing to the considerable content of crude proteins, minerals, polysaccharides, lipids, and other bioactive constituents, which are important for various physiological functions [78]. Alongside, Kang et al. concluded that *C. vulgaris* addition to the diet of broilers improved their body weights [79]. This could explain the improving effect of *C. vulgaris* on the hematological indices of rats.

The liver is the organ of drug transformations and is considered the main hub for protein synthesis; therefore, the cellular damage induced by CdCl_2_ could be monitored by measuring the function of hepatic enzymes in the blood serum, including; AST, ALT and ALP besides the total protein and albumin content [80]. Hence, in case of liver damage due to inflammations, necrosis, or bile duct obstruction, liver enzymes are released in the serum or plasma carrying all the physiological information about the health status [81]. Herein, CdCl_2_ induced liver enzyme synthesis while decreasing the albumin and total protein content compared to the control indicating liver dysfunctions [17,82]. Our results agree with Renugadevi and Prabu, who observed that the level of these enzymes was increased significantly in Cd-intoxicated rats relative to the control [20]. Collectively, the up-regulation of hepatic markers in the blood serum suggested extensive liver injuries in the presence of Cd due to the increased lipid peroxidation that caused membrane damage and increased the membrane’s permeability and the leakage of hepatic enzymes into the circulation [83]. This is in line with the present findings, where the administration of CdCl_2_ significantly increased ROS production and lipid peroxidation while decreasing the TAC in the exposed rat.

Normally, there is a balance between ROS and cellular antioxidants. Hence, oxidative stress occurs if this balance is disturbed by the overproduction of ROS or/and depletion of antioxidants. Subsequently, increasing ROS production can disturb the cell’s physiology and induce DNA, protein, and lipid malformations [84,85]. Moreover, the present findings revealed significant alterations in the lipid profile of Cd-exposed rats relative to the control. Herein, the altered hepatic lipogenesis could be explained by the deleterious effects of Cd on liver functions and structure, which agreed with Wu et al. [86].

The administration of Cd in this study resulted in hepatocyte degeneration, inflammatory cellular infiltration, and blood congestion in the portal area. This is similar to the chronic and acute effects of Cd reported in previous works [82]. Furthermore, hepatic necrosis has been reported in rats after parenteral administration of soluble salt of Cd [87]. In concurrent administration of *C. vulgaris* and Cd, the levels of liver enzyme activity were significantly reduced compared to the Cd group alone. This finding indicates the palliative effects of *C. vulgaris* in ameliorating the hepatotoxic effect of Cd.

Regarding kidney functions, the changes in creatinine and urea levels in the present work showed the adverse effects of CdCl_2_ on renal function, which agreed with Abdel-Moneim and Ghafeer [17]. It has been documented that Cd induced nephrotoxicity via the cross-talk between the liver and kidney via the hepatic Cd-metallothionein complex (Cd-Mt) [19]. For more clarification, the insufficient synthesis of Mt, the unbound Cd ions resulting in hepatic injuries, and the bound ions excreting through the kidney lead to improper function of the renal and lipid peroxidation [20,88,89]. This is parallel to our results concerning the oxidative damage and histopathological alternation in renal tissues, including renal edema [90], proximal tubular apoptosis, necrosis, and degenerations [91] of glomerular capillaries in favor of Bowman’s space [92].

In our present study, Cd induced stressful hormone synthesis suggesting the ability of Cd to activate the hypothalamus-pituitary axis (HPA) and consequently, the release of glucocorticoids from the adrenals served as a significant monitor in estimating the immediate physiological responses to various stressors [93]. On the other hand, administration of *C. vulgaris* with Cd could return the level of these hormones to normal values suggesting its importance as a naturally occurring anti-stress agent [94].

Nuclear factor kB (NF-kB) and heat shock protein 70 (HSP70) are important proteins that help the stability of DNA and organize the transcription processes that protect the cell during stress [95]. In this work, Cd significantly increases the expressions of the NF-kB and HSP70 in the liver, suggesting its significant role in improving the survivability and tolerance of cells against Cd-induced oxidative damage. Contrarily, *C. vulgaris* additions down-regulate the relative expressions of these genes, which may be ascribed to the antioxidant property of *C. vulgaris* [96]. Additionally, NF-κB has a strategic position at the crossroads between oxidative stress and inflammation; it was suggested that ROS might act as a key secondary mediator responsible for the NF-κB activation in response to multiple stimuli [25].

Moreover, the transcription factor, Nrf2, can assist in preventing oxidative damage to cells by maintaining cellular redox homeostasis and promoting the activities of detoxification and biotransformation enzymes [97,98]. Additionally, Nrf2 is essential for producing HO-1, which protects and neutralizes free radicals in cells under stress [99]. Herein, Cd administration downregulated the expressions of Nrf2 and HO-1, indicating the hepatotoxic and oxidative damaging effects of CdCl_2_. However, co-administration of *C. vulgaris* with Cd substantially increased the expressions of HO-1 and Nrf-2, indicating that *C. vulgaris* was successful in triggering the Nrf2/HO-1-dependent pathway against cellular damage caused by CdCl_2_ intoxication.

Regarding the inflammation pathway, the current work studied the activation effect of CdCl_2_ on inflammatory markers, including TNF-α and IL-1β. It may be linked to the enhanced ROS production that occurs after Cd administration and helps recruit more inflammatory cells and fibroblasts to the injury site and stimulates the production of certain cytokines such as TNF-α [100]. Sequentially, TNF-α triggers the inductions of other cytokines such as interferon-γ and IL-1β [101]. Like TNF-α and IL-1β, NO has been regarded as a hepatic injury mediator [102]. Elevating the NO content could lead to lipid peroxidation and more destructive effects on the tissues [103]. Additionally, some research has linked the HSP70 to the induction of TNF-α, IL-1β, and IL-6, which are crucial in the initial stages of liver regeneration [104]. Alongside, HSP 70 has also been demonstrated to be involved in the in vitro induction of NO in bone marrow macrophages [105], and this is consistent with the elevated expressions of HSP 70 in Cd-exposed rats in our study.

In our study, *C. vulgaris* exhibited an excellent hepatoprotective effect, maintained the integrity of membranes of hepatocytes, prevented the leakage of enzymes into the circulation, and repaired the hepatic tissue damage after Cd administration. These results agree with [106,107]. Furthermore, Cheng et al. recorded that the underlined mechanisms of *C. vulgaris* protections might be associated with its immunomodulation activity that could stimulate macrophages’ phagocytic activities and enhance natural killer (NK) cells [108]. *C. vulgaris* microalga has anti-inflammatory and antioxidant properties that protect against membrane fragility [31,32]. These beneficial antioxidants include chlorophyll, carotenoids, astaxanthin, lutein, and phycobili-proteins [109]. The antioxidant properties of *C. vulgaris* and its phenolics provide cellular protective effects due to their redox potentials that could suppress oxygen and decomposing peroxides [110,111,112]. Herein, the protective effect of *C. vulgaris* co-administered with Cd has been elucidated

However, pretreatment of rats with *C. vulgaris* alga, alongside its health benefits, can prevent damage and protect against oxidative harmful effects induced by paracetamol through their free radical scavenging and powerful antioxidant effects, and they can be used as prophylactic agents against paracetamol-induced toxicity [113]. Therefore, there is a possible need for a future investigation on the preventive effect of *C. vulgaris* when administered before exposure to Cd and the curative and restoring effects when administered following damage induction by Cd.

## 5. Conclusions

This study showed that cadmium chloride altered the physiological response, including hematological and serum biochemical parameters (liver and kidney function markers), immunological and inflammatory biomarkers, and induced oxidative stress and histopathological alterations in the liver and kidney of adult male rats. On the other hand, *C. vulgaris* succeeded in preventing the disruption of organ functions by protecting them from oxidative stress and inflammation and enhancing immunity. These effects of *C. vulgaris* could be the mechanisms of their hepato-renal protection. Moreover, the beneficial role of *C. vulgaris* in this study could be suggestive of their use as an immunomodulatory and antioxidant supplement and could be a base for further research on its importance in detoxifying the body from environmental pollutants.

## Figures and Tables

**Figure 1 biomedicines-11-02414-f001:**
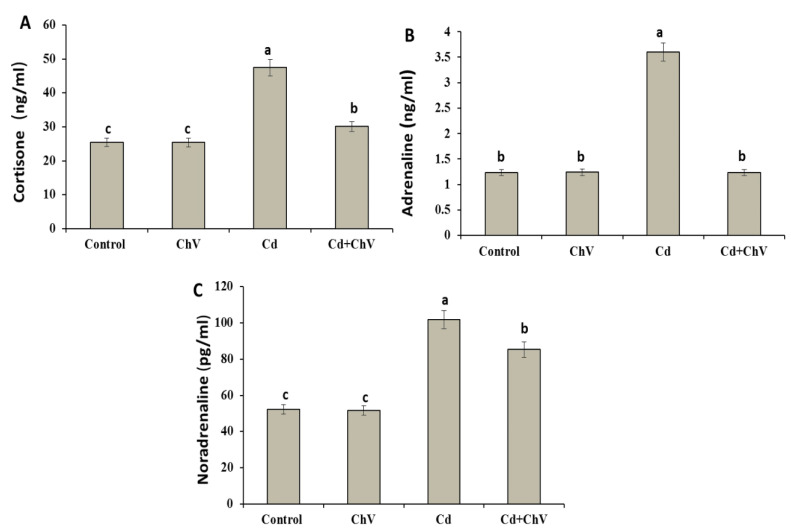
Serum levels of cortisone (**A**), adrenaline (**B**), and noradrenaline (**C**) in rats exposed to Cd and *C. vulgaris* separately or in combination. Data were analyzed using One-way ANOVA followed by Duncan’s Multiple Range test to compare mean value between groups (control, ChV: chlorella vulgaris-treated group, Cd: cadmium-treated group, Cd + ChV: cadmium + *C. vulgaris* treated group). Data were expressed as mean ± SE. Values not sharing a common superscript letter (a, b, c where a: the highest value, c: the lowest value) differ significantly at *p* < 0.05.

**Figure 2 biomedicines-11-02414-f002:**
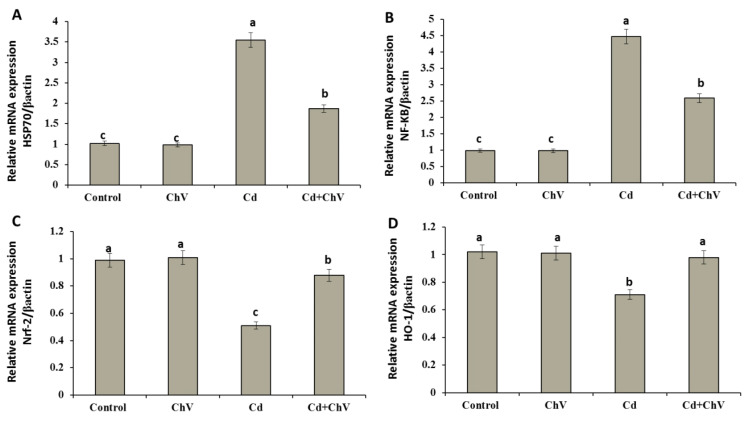
Relative expression of HSP70 (**A**), NF-kB (**B**), Nrf-2 (**C**), or HO-1 (**D**) mRNA in hepatic tissue of rats exposed to Cd and *C. vulgaris* separately or in combination. Data were analyzed using One-way ANOVA followed by Duncan’s Multiple Range test to compare mean value between groups (control, ChV: chlorella vulgaris-treated group, Cd: cadmium-treated group, Cd + ChV: cadmium + *C. vulgaris* treated group). Data were expressed as mean ± SE. Values not sharing a common superscript letter (a, b, c where a: the highest value, c: the lowest value) differ significantly at *p* < 0.05.

**Figure 3 biomedicines-11-02414-f003:**
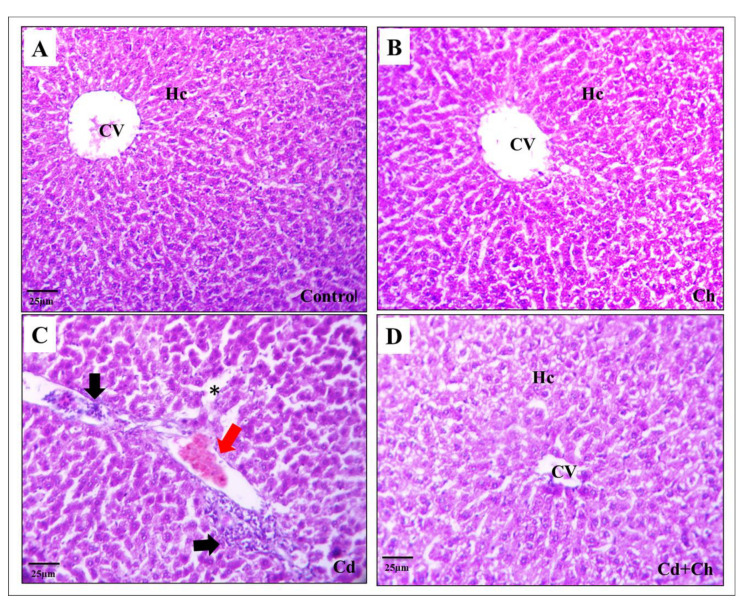
Cadmium alters hepatic tissue architecture: (**A**) Photomicrograph of a hepatic tissue section of the control group stained with H&E showing normal histology with a lobular area of central vein (CV) and hepatocytes (Hc). (**B**) *C. vulgaris*-fed group showing no mild difference from the control group. (**C**) Cadmium treated group showing congestion of the central vein and portal area with inflammatory cellular infiltration (black arrow) and blood congestion (red arrow). Notice the dilated sinusoidal spaces with hepatocyte damage (black asterisk). (**D**) Treated group of cadmium and *C. vulgaris* showing the relief effect of chlorella against the toxic effect of cadmium on the central vein and hepatic cords.

**Figure 4 biomedicines-11-02414-f004:**
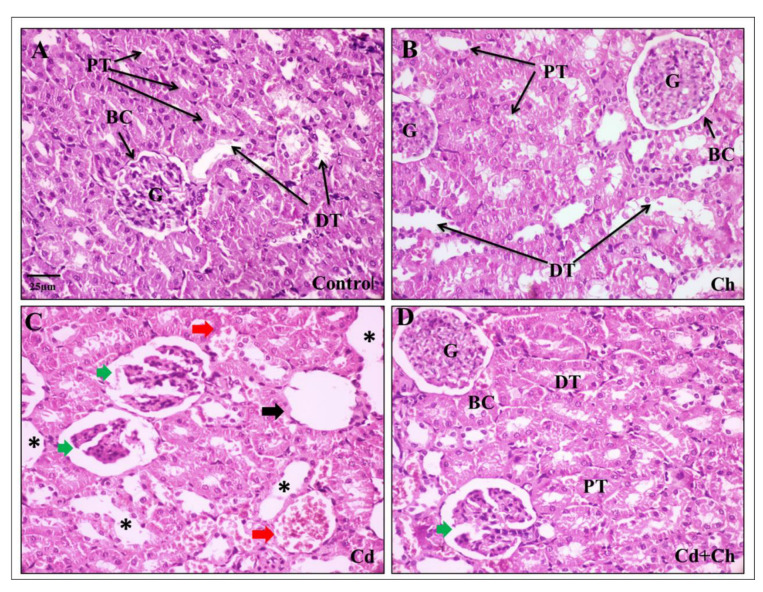
Effects of cadmium and chlorella on the kidney: (**A**) Photomicrograph of control kidney showing normal histology of Bowman capsule (BC), glomerulus (G), proximal convoluted tubule (PT), and distal convoluted tubule (DT). (**B**) Photomicrograph of chlorella-fed group showing normal renal corpuscles. (**C**) Cadmium treated group showing severe degenerative alternations in the renal tubules (black asterisk), corpuscle degeneration (black arrow), glomeruli segmentation with Bowman space widening (green arrow), and blood congestion (red arrow). (**D**) Cd + *C. vulgaris* co-administrated group showing the protective change of *C. vulgaris* on the glomeruli and tubules epithelial cells except for mild corpuscle segmentation (green arrow).

**Table 1 biomedicines-11-02414-t001:** Effects of separate and concurrent exposure to CdCl_2_ and ChV on hematological parameters.

	Control	ChV	Cd	Cd + ChV	*p* Value
RBCs (10^6^/mm^3^)	8.35 ± 0.12	8.37 ± 0.12	7.21 ± 0.56	8.06 ± 0.06	0.074
Hb (gm/dL)	13.38 ± 0.35 ^a^	13.24 ± 0.16 ^a^	8.68 ± 0.29 ^b^	12.44 ± 0.12 ^a^	<0.001
PCV %	50.39 ± 0.17 ^a^	50.91 ± 0.94 ^a^	43.44 ± 0.51 ^c^	46.66 ± 0.40 ^b^	<0.001
WBC (10^3^/mm^3^)	8.52 ± 0.07 ^a^	8.54 ± 0.09 ^a^	6.21 ± 0.01 ^c^	7.20 ± 0.13 ^b^	<0.001
Lymphocyte (10^3^/mm^3^)	4.34 ± 0.23 ^a^	4.69 ± 0.07 ^a^	3.38 ± 0.13 ^b^	4.18 ± 0.14 ^a^	0.002
Neutrophil (10^3^/mm^3^)	2.71 ± 1.35	4.06 ± 0.01	4.02 ± 0.01	4.04 ± 0.02	0.452
Esinophil (10^3^/mm^3^)	0.82 ± 0.01	0.83 ± 0.01	0.82 ± 0.01	0.82 ± 0.01	0.583

Data were analyzed using One-way ANOVA followed by Duncan’s Multiple Range test to compare mean value between groups (control, ChV: chlorella vulgaris-treated group, Cd: cadmium-treated group, Cd + ChV: cadmium + *C. vulgaris* treated group). Data were expressed as mean ± SE. Values not sharing a common superscript letter (a, b, c where a: the highest value, c: the lowest value) differ significantly at *p* < 0.05.

**Table 2 biomedicines-11-02414-t002:** Effect of separate and concurrent exposure to CdCl_2_ and *Chlorella* on body weight and other organs.

	Control	ChV	Cd	Cd + ChV	*p* Value
Initial weight	185.31 ± 0.08	185.29 ± 0.09	185.30 ± 0.08	185.30 ± 0.07	0.997
Final weight	209.59 ± 0.47 ^b^	215.55 ± 1.70 ^a^	200.84 ± 0.41 ^c^	209.14 ± 0.12 ^b^	<0.001
Relative liver weight	3.30 ± 0.01	3.30 ± 0.01	3.29 ± 0.01	3.29 ± 0.02	0.802
Relative kidney weight	0.79 ± 0.02	0.78 ± 0.02	0.79 ± 0.02	0.78 ± 0.03	0.995
Relative spleen weight	0.38 ± 0.01	0.36 ± 0.01	0.38 ± 0.01	0.37 ± 0.01	0.640

Data were analyzed using One-way ANOVA followed by Duncan’s Multiple Range test to compare mean value between groups (control, ChV: *C. vulgaris*-treated group, Cd: cadmium-treated group, Cd + ChV: cadmium + *C. vulgaris* treated group). Data were expressed as mean ± SE. Values not sharing a common superscript letter (a, b, c where a: the highest value, c: the lowest value) differ significantly at *p* < 0.05.

**Table 3 biomedicines-11-02414-t003:** Effects of separation and concurrent exposure to CdCl_2_ and *Chlorella* on serum biochemical parameters.

	Control	ChV	Cd	Cd + ChV	*p* Value
Liver function markers					
Total protein(g/dL)	8.04 ± 0.04 ^a^	8.12 ± 0.01 ^a^	5.05 ± 0.03 ^c^	7.28 ± 0.15 ^b^	<0.001
Albumin(g/dL)	5.01 ± 0.01 ^a^	5.08 ± 0.06 ^a^	2.52 ± 0.01 ^c^	4.84 ± 0.04 ^b^	<0.001
ALT(U/L)	28.35 ± 0.12 ^c^	28.16 ± 0.02 ^c^	121.17 ± 0.07 ^a^	82.87 ± 03.68 ^b^	<0.001
AST(U/L)	67.45 ± 0.41 ^c^	66.55 ± 01.01 ^c^	151.44 ± 01.32 ^a^	113.90 ± 03.12 ^b^	<0.001
ALP(U/L)	112.53 ± 0.25 ^c^	110.38 ± 0.08 ^c^	196.81 ± 01.14 ^a^	150.24 ± 05.51 ^b^	<0.001
Kidney function markers					
Urea(mg/dL)	18.75 ± 0.12 ^c^	17.21 ± 0.01 ^c^	41.32 ± 0.61 ^a^	23.75 ± 1.21 ^b^	<0.001
Creatinine (mg/dL)	0.69 ± 0.00 ^b^	0.68 ± 0.01 ^b^	1.01 ± 0.01 ^a^	0.69 ± 0.01 ^b^	<0.001
Lipid profile					
Triglycerides (mg/dL)	100.20 ± 0.42	98.62 ± 0.69	105.39 ± 0.35	103.97 ± 01.26	0.447
Total cholesterol (mg/dL)	85.97 ± 0.92 ^c^	85.58 ± 0.32 ^c^	157.35 ± 0.82 ^a^	122.47 ± 1.52 ^b^	<0.001
HDL-cholesterol (mg/dL)	50.28 ± 0.04 ^a^	50.12 ± 0.05 ^ab^	30.29 ± 0.09 ^c^	49.36 ± 0.41 ^b^	<0.001
LDL-cholesterol(mg/dL)	30.21 ± 0.03 ^c^	30.18 ± 0.03 ^c^	122.59 ± 1.20 ^a^	70.35 ± 0.11 ^b^	<0.001

Data were analyzed using One-way ANOVA followed by Duncan’s Multiple Range test to compare mean value between groups (control, ChV: *C. vulgaris*-treated group, Cd: cadmium-treated group, Cd + ChV: cadmium + *C. vulgaris* treated group). Data were expressed as mean ± SE. Values not sharing a common superscript letter (a, b, c where a: the highest value, c: the lowest value) differ significantly at *p* < 0.05.

**Table 4 biomedicines-11-02414-t004:** Effects of separate and concurrent exposure to CdCl_2_ and *Chlorella* on oxidative stress, immunity, and inflammatory markers in serum.

	Control	ChV	Cd	Cd + V	*p* Value
Oxidative stress markers					
TAC (μmol/mL)	3.46 ± 0.13 ^b^	4.14 ± 0.12 ^a^	1.02 ± 0.01 ^d^	2.87 ± 0.01 ^c^	<0.001
ROS (U/mL)	15.55 ± 0.10 ^c^	15.85 ± 0.09 ^c^	30.24 ± 0.02 ^a^	18.30 ± 0.06 ^b^	<0.001
MDA (nmol/mL)	5.01 ± 0.01 ^b^	4.63 ± 0.20 ^b^	13.49 ± 0.58 ^a^	5.03 ± 0.01 ^b^	<0.001
Immunity biomarkers					
IgG (mg/dL)	8.21 ± 0.01 ^b^	8.73 ± 0.09 ^a^	5.26 ± 0.10 ^c^	8.20 ± 0.01 ^b^	<0.001
IgM (mg/dL)	1.33 ± 0.07	1.39 ± 0.11	1.30 ± 0.06	1.31 ± 0.06	0.843
Inflammatory markers					
TNF-α (pg/mL)	63.26± 1.54 ^c^	60.06 ± 0.10 ^c^	99.14 ± 0.27 ^a^	75.20 ± 2.51 ^b^	<0.001
IL-1β (pg/mL)	72.02 ± 0.24 ^b^	70.17 ± 0.10 ^c^	131.04 ± 0.55 ^a^	131.04 ± 0.55 ^a^	<0.001
NO (Umol/L)	51.19 ± 1.01 ^c^	50.13 ± 0.07 ^c^	81.83 ± 1.13 ^a^	56.38 ± 0.45 ^b^	<0.001

Data were analyzed using One-way ANOVA followed by Duncan’s Multiple Range test to compare mean value between groups (control, ChV: *C. vulgaris*-treated group, Cd: cadmium-treated group, Cd + ChV: cadmium + *C. vulgaris* treated group). Data were expressed as mean ± SE. Values not sharing a common superscript letter (a, b, c where a: the highest value, c: the lowest value) differ significantly at *p* < 0.05.

## Data Availability

The data presented in this study are available on request from the corresponding authors.

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
