# Peer review of "Benefits of Chlorella vulgaris against Cadmium Chloride-Induced Hepatic and Renal Toxicities via Restoring the Cellular Redox Homeostasis and Modulating Nrf2 and NF-KB Pathways in Male Rats"

_biomedicines, 2023, doi:10.3390/biomedicines11092414_

Round 1

Reviewer 1 Report (New Reviewer)

1.    English style and grammar need to improve

2.    In the abstract, add the time of experiment (injection days).

3.    The novelty of the study also should be presented at the end of the Intro

4.    Line 106 “BW”, at the first time, the names should be full-written without abbreviation

5.    Material and Methods is very poor, missed several important details, and need to be extensively improved:

-        Lines 110-113: Provide more information about CdCl2, like conc., commercial or lab grade, ….

-        2.4. Blood Collecting and Tissue Preservation, 2.5. Hematological Analysis, and 2.6. Biochemical Analysis: How many samples from each treatment? What references?

-         2.7, 2.8, and 2.9: What references?

-        Line 182: mean ± SEM “SEM” or “SE”

6.      Line 1184: “3.1. Chlorella v. restores Hematological Parameters Altered by Cadmium”. Firstly the subtitle needs to be rewritten and improve. On other important point, I don’t now what authors would to say “Chlorella v.” Here in this stydy, the full scientific name for Chlorella is “Chlorella vulgaris”. The authors may write it  "Chlorella vulgaris” or C. vulgaris” but not Chlorella v.” !!!!!!. Please, revise it in all manuscript sections.

7.      Table 3, 4, and 5: correct scientific name to Chlorella vulgaris not Chlorella

8.      Again, in all manuscript section, revise the scientific name, there are many errors like: chlorella vulgaris “c”, regular not etalic,  Chlorella v.

9.      The conclusions need further improvements. It should build on what is already there to say something about the significance of the findings for existing research, future research, and study implications.

Extensive editing of English language required

Author Response

Reviewer 1

  1. English style and grammar need to improve

Response : done..Thank you very much

  1. In the abstract, add the time of experiment (injection days).

Response: added

  1. The novelty of the study also should be presented at the end of the Intro

Response: added

  1. Line 106 “BW”, at the first time, the names should be full-written without abbreviation

Response: done

  1. Material and Methods is very poor, missed several important details, and need to be extensively improved:

-        Lines 110-113: Provide more information about CdCl2, like conc., commercial or lab grade, ….

Response : cadmium chloride was in the form of white powder of  analytical grade (99%) purity.

-        2.4. Blood Collecting and Tissue Preservation, 2.5. Hematological Analysis, and 2.6. Biochemical Analysis: How many samples from each treatment? What references?

we used 5 samples/group and conducted the biochemical analysis following the protocol of the kits manufacturer and the references were added as suggested.

-         2.7, 2.8, and 2.9: What references?

Response : added

-        Line 182: mean ± SEM “SEM” or “SE”

Response : corrected

  1. Line 1184: “3.1. Chlorella v. restores Hematological Parameters Altered by Cadmium”. Firstly the subtitle needs to be rewritten and improve. On other important point, I don’t now what authors would to say “Chlorella v.” Here in this stydy, the full scientific name for Chlorella is “Chlorella vulgaris”. The authors may write it  "Chlorella vulgaris” or C. vulgaris” but not Chlorella v.” !!!!!!. Please, revise it in all manuscript sections.

Response : done

  1. Table 3, 4, and 5: correct scientific name to Chlorella vulgaris notChlorella

Response: done

  1. Again, in all manuscript section, revise the scientific name, there are many errors like: chlorella vulgaris “c”, regular not etalic,  Chlorellav.

Response: done

  1. The conclusions need further improvements. It should build on what is already there to say something about the significance of the findings for existing research, future research, and study implications.

Response: done

Reviewer 2 Report (New Reviewer)

In this paper the Authors demonstrate the effects of the green microalga Chlorella vulgaris (C. vulgaris) against the hepatic and renal toxicity induced by cadmium in adult male rats.

Overall the paper is well written and the rationale is fine. The data lend support to the hypothesis that Chlorella v might represent a new tool to counteract the Cd-induced damage. However some changes are needed before publication.

 In the Introduction, regarding the cytotoxic effects of cadmium and its ability to induce an oxidative damage, the authors have left out some recent citations. Please see and comment doi: 10.3389/fcell.2020.604377.

Discussion section: the discussion, despite being well written and articulate, is perhaps a little too long. I would shorten it by eliminating the parts concerning the effects of treatment with cadmium alone (already widely supported by the literature).

Furthermore, the authors decided to administer Chlorella v in co-treatment with cadmium and demonstrated its effectiveness in counteracting the toxic effects. Would the alga have an effect even if administered as a pre-treatment? The authors should add a comment on this important aspect under the Discussion section. 

Author Response

Reviewer 2

in this paper the Authors demonstrate the effects of the green microalga Chlorella vulgaris (C. vulgaris) against the hepatic and renal toxicity induced by cadmium in adult male rats.

Overall the paper is well written and the rationale is fine. The data lend support to the hypothesis that Chlorella v might represent a new tool to counteract the Cd-induced damage. However some changes are needed before publication.

Response: thank you very much for this supportive comment

 In the Introduction, regarding the cytotoxic effects of cadmium and its ability to induce an oxidative damage, the authors have left out some recent citations. Please see and comment doi: 10.3389/fcell.2020.604377.

Response: added

Discussion section: the discussion, despite being well written and articulate, is perhaps a little too long. I would shorten it by eliminating the parts concerning the effects of treatment with cadmium alone (already widely supported by the literature).

response: we improved as suggested

Furthermore, the authors decided to administer Chlorella v in co-treatment with cadmium and demonstrated its effectiveness in counteracting the toxic effects. Would the alga have an effect even if administered as a pre-treatment? The authors should add a comment on this important aspect under the Discussion section. 

Response: thanks a lot, added to the discussion as suggested

Reviewer 3 Report (New Reviewer)

The paper: “Benefits of Chlorella vulgaris Against Cadmium Chloride-Induced Hepatic and Renal Toxicities via Restoring the Cellular Redox Homeostasis and Modulating Nrf2 and NF-KB pathways in Male Rats” presents an interesting work about the preventive effects of Chlorella vulgaris against toxicity of cadmium in liver and kidney. The paper is well written, images good and results convincing. Only minor remarks, detailed below, need attention to make the paper suitable for publication.

Lines 116-119- the administration procedure for Chlorella vulgaris (500 mg/kg/day for 10 days) should be clearly given. Also, group 4 is reported to have been co-treated with both 2 mg/kg CdCl2 and 500 mg/kg Chlorella vulgaris daily for 10 days. Hence, Chlorella vulgaris treatment can be considered as protective against Cd induction of damages. In this respect, not mandatory but as a comment  about a possible limitation of this study, a sentence could be added, for example to the final part of Discussion, on a possible need for a future investigation on curative, restoring effects  of Chlorella vulgaris when administered following damage induction by Cd.

Line 153: “..total antioxidant capacity (TAC)” and Table 5: “(nmol of Trolox equivalent/l)”- the rationale of this assay should be better elucidated.

Line 200, and similar sentences in next Figure legends - The sentence “Values are not sharing a common superscript letter (a, b, c where a: the highest value, c: the lowest value) differ significantly at p < 0.05” is to be revised, for example: “Values not sharing a common superscript letter (a, b, c where a: the highest value, c: the lowest value) differ significantly at p < 0.05”; or :” Values which are not sharing a common superscript letter (a, b, c where a: the highest value, c: the lowest value) differ significantly at p < 0.05”. Is it?

A final comment about the Title of the Special Issue: “The Role of Microglia in Human Diseases” – how can the Authors justify the fit of this good work to the special issue?

good

Author Response

Reviewer 3

The paper: “Benefits of Chlorella vulgaris Against Cadmium Chloride-Induced Hepatic and Renal Toxicities via Restoring the Cellular Redox Homeostasis and Modulating Nrf2 and NF-KB pathways in Male Rats” presents an interesting work about the preventive effects of Chlorella vulgaris against toxicity of cadmium in liver and kidney. The paper is well written, images good and results convincing. Only minor remarks, detailed below, need attention to make the paper suitable for publication.

Response: thank you very much for this supportive comment

Lines 116-119- the administration procedure for Chlorella vulgaris (500 mg/kg/day for 10 days) should be clearly given. Also, group 4 is reported to have been co-treated with both 2 mg/kg CdCl2 and 500 mg/kg Chlorella vulgaris daily for 10 days. Hence, Chlorella vulgaris treatment can be considered as protective against Cd induction of damages. In this respect, not mandatory but as a comment  about a possible limitation of this study, a sentence could be added, for example to the final part of Discussion, on a possible need for a future investigation on curative, restoring effects  of Chlorella vulgaris when administered following damage induction by Cd.

Response: added, and the discussion and the conclusion is also improved

Line 153: “..total antioxidant capacity (TAC)” and Table 5: “(nmol of Trolox equivalent/l)”- the rationale of this assay should be better elucidated.

Response: done

Line 200, and similar sentences in next Figure legends - The sentence “Values are not sharing a common superscript letter (a, b, c where a: the highest value, c: the lowest value) differ significantly at p < 0.05” is to be revised, for example: “Values not sharing a common superscript letter (a, b, c where a: the highest value, c: the lowest value) differ significantly at p < 0.05”; or :” Values which are not sharing a common superscript letter (a, b, c where a: the highest value, c: the lowest value) differ significantly at p < 0.05”. Is it?

Response: done

A final comment about the Title of the Special Issue: “The Role of Microglia in Human Diseases” – how can the Authors justify the fit of this good work to the special issue?

Response: Because Chlorella sp. supplementation revealed benefcial physiological effects such as   antihypertensive,   antoxidative,   hypocholesterolemic, and antitumor   activities, hypoglycemic and hypolipidemic   effects in animal and human studies. 

Round 2

Reviewer 1 Report (New Reviewer)

Accept in present form

Author Response

Comments: 1. Please relocate Table 1 to the supplementary data and also
include information about the annealing temperature."

Au: done 

  1. Please specify that the biomarkers were detected in the serum (Table 5).

Au: done 

3.The units for IgG and IgM are missing (Table 5).

Au: added 

  1. Why is the unit for serum TAC given as umol/g instead of umol/ml?"

Au: corrected

  1. Only a single dose was used in this study, and there was no positive
    control either.

Au: We used 4 groups  : 1. control (untreated)

  1. C.vulgaris group (positive control)
  2. Cadmium treated group (intoxicated group)
  3. Cadmium+C.vulgaris group

This manuscript is a resubmission of an earlier submission. The following is a list of the peer review reports and author responses from that submission.

Round 1

Reviewer 1 Report

The research article entitled " Benefits of Chlorella vulgaris Against Cadmium Chloride-Induced Hepatic and Renal Toxicities via Restoring the Cellular 3 Redox Homeostasis and Modulating Nrf2 and NF-KB pathways 4 in Male Rats” described the hepatoprotective and renal protective effect of Chrlorella vulgaris (Cv) on Cadmium (Cg) included toxicities in rats. Author designed 10 days experimental study with appropriate read out. Author demonstrated the Cv efficacy in haematological, biochemical (liver and kidney function test), immunological and histological parameters. Author is hypothesized that the efficacy of the Cv is mediated via modification of NFkB pathway. The experimental design is quite appropriate with impressive results. Author can mention about the role of NFkB in  oxidative stress and its implication in various inflammatory disease condition with supporting articles citation. Over in all, the article is narrating the Cv efficacy in Cd ind-toxicity model with substantiated data, however the following clarification needs to be addressed.

1.      WBC levels were expressed as cell count. The differential count (DC) percentage can express clearly which cell types differentiated or involved in the Cd induced toxicity. Can it possible to include percentage distribution of DC.

2.      On each table, method of statistical analysis was not mentioned. Superscriptions (a, b, c) stand for what? Can author state clearly about comparison of groups? From the statement, we could not understand that what are the groups are compared.

3.      Cd reduced the IgG level while IgM levels was unchanged, Any explanation for these phenomena.

4.      In Figures, what type of statistical analysis was performed? Please state clearly the comparative groups for a,b,c.

5.      The oxidative stress is mainly assessed via GSH and GSSH levels by most of the researcher. Author could have explained this study with GSH/GSSG ratio from the tissue homogenate.

6.      Author did not state study termination day in the experimental design section. Please include details.

7.      Cd reduced the body weight gain by ~4% compared to control group, whereas relative liver and kidney weight did not alter compared to control group, which indirectly saying that Cd reduces body weight as well organ weight. Usually, any toxic materials increase the inflammation on any organs leads to increases the organ weight (at inflammatory stage) but here organ weight is unchanged in Cd control group. Any explanation for this?

8.      How was the relative liver, spleen and kidney weight calculated? Can explain with more clarity (formula).

9.      The author is screening Chlorella vulgaris for the hepatoprotective or detoxifications property.  To validate the Chlorella vulgaris, addition of Silymarin (current standard care for hepatotoxicity) (Silymarin) treatment groups would have helpful.

10.  Author is mainly substantiating the efficacy of Cv via NfkB pathway, so we would suggest to include role of Nfkb in oxidative stress with supporting article and NfKB implication on various inflammatory diseases on introduction and discussion sections

TThanks.

Need to be improved in some sentences, Over in all, content is fine and conveying the message

Author Response

Reviewer 1

The research article entitled " Benefits of Chlorella vulgaris Against Cadmium Chloride-Induced Hepatic and Renal Toxicities via Restoring the Cellular 3 Redox Homeostasis and Modulating Nrf2 and NF-KB pathways 4 in Male Rats” described the hepatoprotective and renal protective effect of Chrlorella vulgaris (Cv) on Cadmium (Cg) included toxicities in rats. Author designed 10 days experimental study with appropriate read out. Author demonstrated the Cv efficacy in haematological, biochemical (liver and kidney function test), immunological and histological parameters. Author is hypothesized that the efficacy of the Cv is mediated via modification of NFkB pathway. The experimental design is quite appropriate with impressive results.. Author can mention about the role of NFkB in  oxidative stress and its implication in various inflammatory disease condition with supporting articles citation. Over in all, the article is narrating the Cv efficacy in Cd ind-toxicity model with substantiated data, however the following clarification needs to be addressed.

  1. WBC levels were expressed as cell count. The differential count (DC) percentage can express clearly which cell types differentiated or involved in the Cd induced toxicity. Can it possible to include percentage distribution of DC.

Response: thank you very much for this notice, we add it as suggested.

  1. On each table, method of statistical analysis was not mentioned. Superscriptions (a, b, c) stand for what? Can author state clearly about comparison of groups? From the statement, we could not understand that what are the groups are compared.

Response: Data were analyzed using one way-ANOVA procedure.  Duncan’s Multiple Range test was conducted to compare means value between groups (control, ChV: chlorella vulgaris-treated group, Cd:cadmium-treated group, Cd+ChV: cadmium+chlolerlla vulgaris treated group) , . Data were expressed as mean ± SE .Values are not sharing a common superscript letter (a, b, c where a: the highest value, c: the lowest value) differ significantly at p < 0.05.

  1. Cd reduced the IgG level while IgM levels was unchanged, Any explanation for these phenomena.

Response: each class represents a group of antibodies and has a slightly different role, normally; IgG is the most abundant antibody in the blood (About 70-80% of the immunoglobulins in the blood). A variety of conditions can cause an increase (hypergammaglobulinemia) or decrease (hypogammaglobulinemia) in the production of immunoglobulins. Some cause an excess or deficiency of all classes of immunoglobulins while others affect only one class.

Some possible causes of low levels of one or more immunoglobulins are: Conditions that may reduce the amount of protein in your body, including:Kidney disease, Serious burns, Certain malabsorption disorders, Malnutrition

  1. In Figures, what type of statistical analysis was performed? Please state clearly the comparative groups for a, b, c.

Response : added

  1. The oxidative stress is mainly assessed via GSH and GSSH levels by most of the researcher. Author could have explained this study with GSH/GSSG ratio from the tissue homogenate.

Response:   in this study we did not measure individual antioxidants but we measured the total antioxidant capacity. and the other individual antioxidants were added In a second study which is still under publication so we tried to avoid the repletion of data. We used some other measures of oxidative stress such as reactive oxygen species (ROS), and lipid peroxidation (MDA)  

  1. Author did not state study termination day in the experimental design section. Please include details.

Response: the experiment lasted for 10 days, we clarified this in the material as suggested .

  1. Cd reduced the body weight gain by ~4% compared to control group, whereas relative liver and kidney weight did not alter compared to control group, which indirectly saying that Cd reduces body weight as well organ weight. Usually, any toxic materials increase the inflammation on any organs leads to increases the organ weight (at inflammatory stage) but here organ weight is unchanged in Cd control group. Any explanation for this?

Response: the weight of organs in this experiment was related to the body weight of their correspondence animals in each group so they might not change than control .

  1. How was the relative liver, spleen and kidney weight calculated? Can explain with more clarity (formula).

Response: added ( it means the weight of the organ on the weight of animal multiplied in 100 to get the percentage of organ weight (organ weight x100 / body weight )

  1. The author is screening Chlorella vulgaris for the hepatoprotective or detoxifications property.  To validate the Chlorella vulgaris, addition of Silymarin (current standard care for hepatotoxicity) (Silymarin) treatment groups would have helpful.

Response: thank you very much for this useful suggestion; we will consider this in a new research to confirm the present work as suggested.

  1. Author is mainly substantiating the efficacy of Cv via NfkB pathway, so we would suggest to include role of Nfkb in oxidative stress with supporting articleand NfKB implication on various inflammatory diseases on introduction and discussion sections

Response: Added

Reviewer 2 Report

In the manuscript „Benefits of Chlorella vulgaris Against Cadmium Chloride Induced Hepatic and Renal Toxicities via Restoring the Cellular Redox Homeostasis and Modulating Nrf2 and NF-KB pathways in Male Rats “, the authors present hepatorenal protective effect of Chlorella vulgaris. The authors measured various biochemical parameters, joined with various laboratory test analyses. They also determined the markers/hormones/neurotransmitters related to oxidative stress, transcriptional level of NRF2 KappaB1, HMOX1 and HSP70. The study also included pathohistological examination of liver and kidney.

They concluded that Chlorella vulgaris indeed has hepatorenal protective effect.

It has been shown long time ago.

Regarding hepatoprotective effect – this is not a novel information. Not cited in the manuscript, in 2008, Jae-Young Shim et al undoubtedly showed the hepatoprotective effect in a very similar experimental model (40 Sprague-Dawley rats.

Shim JY, Shin HS, Han JG, Park HS, Lim BL, Chung KW, Om AS. Protective effects of Chlorella vulgaris on liver toxicity in cadmium-administered rats. J Med Food. 2008 Sep;11(3):479-85. doi: 10.1089/jmf.2007.0075.)

Regarding protective effect on kidney – Shim et al. showed some of effects already in 2009 (Shim JA, Son YA, Park JM, Kim MK. Effect of Chlorella intake on Cadmium metabolism in rats. Nutr Res Pract. 2009 Spring;3(1):15-22. doi: 10.4162/nrp.2009.3.1.15.)

Still, the authors show some interesting data related to transcriptional activity of four genes.

The molecular mechanism of protection associated with Chlorella vulgaris in animals exposed to cadmium does not seem to be shown. That cannot be shown based on transcriptional analysis of four genes. Of importance, the authors did not show modulation of kappaB and Nrf2 related pathways (as stated in the title).

Considering previously shown hepatorenal protective effect of Chlorella vulgaris in previously published studies  which also covered some - but not all biochemical analyses presented in this manuscript, as well as liver histology) the readers may want to see an extension of research that relate to molecular analyses.

The level of transcription does not necessarily indicate the level of the protein. The authors should make the content of the manuscript stronger and perform immunohistochemistry, for confirming the RT-qPCR data. I   even more of importance, because one would like to see the cellular localization of the two transcription factors (is NF-kappaB actually the Nfkb1?).

Since the authors have extremely low and uniformed standard deviations related to RT-qPCR (which is quite unusual when working with damaged tissues), I would like to see the row RT-qPCR data.

Thank you.

No comment.

Author Response

In the manuscript „Benefits of Chlorella vulgaris Against Cadmium Chloride Induced Hepatic and Renal Toxicities via Restoring the Cellular Redox Homeostasis and Modulating Nrf2 and NF-KB pathways in Male Rats “, the authors present hepatorenal protective effect of Chlorella vulgaris. The authors measured various biochemical parameters, joined with various laboratory test analyses. They also determined the markers/hormones/neurotransmitters related to oxidative stress, transcriptional level of NRF2 KappaB1, HMOX1 and HSP70. The study also included pathohistological examination of liver and kidney.

They concluded that Chlorella vulgaris indeed has hepatorenal protective effect.

It has been shown long time ago.

Regarding hepatoprotective effect – this is not a novel information. Not cited in the manuscript, in 2008, Jae-Young Shim et al undoubtedly showed the hepatoprotective effect in a very similar experimental model (40 Sprague-Dawley rats.

Shim JY, Shin HS, Han JG, Park HS, Lim BL, Chung KW, Om AS. Protective effects of Chlorella vulgaris on liver toxicity in cadmium-administered rats. J Med Food. 2008 Sep;11(3):479-85. doi: 10.1089/jmf.2007.0075.)

Regarding protective effect on kidney – Shim et al. showed some of effects already in 2009 (Shim JA, Son YA, Park JM, Kim MK. Effect of Chlorella intake on Cadmium metabolism in rats. Nutr Res Pract. 2009 Spring;3(1):15-22. doi: 10.4162/nrp.2009.3.1.15.)

Response: thank you very much for such valuable references, we used them as suggested. they available to us as abstracts only, so we are sorry for missing them I the manuscript.

Still, the authors show some interesting data related to transcriptional activity of four genes.

The molecular mechanism of protection associated with Chlorella vulgaris in animals exposed to cadmium does not seem to be shown. That cannot be shown based on transcriptional analysis of four genes. Of importance, the authors did not show modulation of kappaB and Nrf2 related pathways (as stated in the title).

Response: Chlorella vulgaris improved the transcription level of kappaB and Nrf2 however they did not reach the control value, it might need more time , so we will consider this in a second study to confirm this one.

Considering previously shown hepatorenal protective effect of Chlorella vulgaris in previously published studies  which also covered some - but not all biochemical analyses presented in this manuscript, as well as liver histology) the readers may want to see an extension of research that relate to molecular analyses.

Response: thank you very much for this useful suggestion; we will consider this in a new research to confirm the present work as suggested.

The level of transcription does not necessarily indicate the level of the protein. The authors should make the content of the manuscript stronger and perform immunohistochemistry, for confirming the RT-qPCR data. I   even more of importance, because one would like to see the cellular localization of the two transcription factors (is NF-kappaB actually the Nfkb1?).

Response: thanks for this suggestion, we totally agree with this, but unfortunately it will require to repeat the experiment to get fresh sections of the tissue and it would costs a lot for obtaining the antibodies. 

Since the authors have extremely low and uniformed standard deviations related to RT-qPCR (which is quite unusual when working with damaged tissues), I would like to see the row RT-qPCR data.

Response: we just delete the standard error as it is not accepted in designing of chars, we culd not add a table and a figure in the same manuscript. The obtained statistical analysis of data in this table 

Control

ChV

CdCl2

CdCl2+ ChV

p value

NF-kB

0.99±0.01c

0.98±0.00c

4.48±0.02a

2.59±0.02b

.000

HSP70

1.02±0.04c

0.98±0.01c

3.55±0.04a

1.87±0.01b

.000

Nrf2

0.99±0.00a

1.01±0.01a

0.51±0.01c

0.88±0.01b

.000

HO-1

1.02±0.04a

1.01±0.00a

0.71±0.01b

0.98±0.01a

.000

Round 2

Reviewer 2 Report

Dear authors,

Thank you for all explanations given.